# Coordinated stimulation of axon regenerative and neurodegenerative transcriptional programs by ATF4 following optic nerve injury

**Preethi Somasundaram[1], Madeline M Farley[1], Melissa A Rudy[1,2], Katya Sigal[2], Andoni I Asencor[2], David G Stefanoff[1], Malay Shah[1], Puneetha Goli[1], Jenny Heo[1], Shufang Wang[1], Nicholas M Tran[3], Trent A Watkins[1,2]\***

[1]Departments of Neurosurgery, Baylor College of Medicine, Houston, United States; [2]Division of Neuroimmunology and Glial Biology, Department of Neurology, University of California, San Francisco, San Francisco, United States; [3]Mol. and Human Genetics, Baylor College of Medicine, Houston, United States

**\*For correspondence:**
trent.watkins@ucsf.edu

**Competing interest:** The authors declare that no competing interests exist.

## eLife Assessment

This study presents a **valuable** finding about the role of Perk (Protein kinase RNA-like endoplasmic reticulum kinase) and Atf4 (Activating Transcription Factor-4) in the integrated neurodegenerative and regenerative responses following the optic nerve injury. The authors present **solid** evidence, combining newly generated transcriptomic data with publicly available datasets to strengthen their findings. Despite some limitations in data quality and interpretation, the study is likely to be of interest to researchers studying optic neuropathies and axonal regeneration.

**Abstract** Stress signaling is important for determining the fates of neurons following axonal insults. Previously, we showed that the stress-responsive kinase PERK contributes to injury-induced neurodegeneration (Larhammar et al., 2017). Here, we show that PERK acts primarily through activating transcription factor-4 (ATF4) to stimulate not only pro-apoptotic but also pro-regenerative responses following optic nerve damage. Using conditional knockout mice, we find an extensive PERK/ATF4-dependent transcriptional response that includes canonical ATF4 target genes and modest contributions by C/EBP Homologous Protein (CHOP). Overlap with c-Jun-dependent transcription suggests interplay with a parallel stress pathway that orchestrates regenerative and apoptotic responses. Accordingly, neuronal knockout of ATF4 recapitulates the neuroprotection afforded by PERK deficiency, and PERK or ATF4 knockout impairs optic axon regeneration enabled by disrupting the tumor suppressor PTEN. These findings reveal an integral role for PERK/ATF4 in coordinating neurodegenerative and regenerative responses to CNS axon injury.

## Introduction

Experiments using conditional knockout mice have revealed a critical role for stress signaling-mediated transcriptional responses in determining the fates of neurons after axon injury. Prominent among these, neuronal knockout of the stress-responsive Dual leucine-zipper kinase (DLK) suppresses the stimulation by injury of transcription factors of the bZIP family, including c-Jun and the activating transcription factor-3 (ATF3), that substantially alter the transcriptomes of distressed neurons (*Asghari Adib et al., 2018*; *Farley and Watkins, 2018*; *Le Pichon et al., 2017*; *Tran et al., 2019*; *Welsbie*

*et al., 2017; Welsbie et al., 2013*). As is common in cellular stress signaling, this injury-induced MAP kinase (MAPK) signaling cascade couples the activation of transcriptional programs that prime damaged neurons for repair with those that prime those same neurons for apoptosis, the latter a feature that may facilitate the elimination of irreparable cells (*Watkins et al., 2013*).

These and other essential insights have been enabled by mouse optic nerve crush, an exceptionally tractable model for CNS axon injury that has been valuable for understanding axonopathy-driven CNS neurodegeneration, particularly glaucoma, and the neuron-intrinsic and extrinsic factors that limit CNS axon regeneration (*Lindborg et al., 2021; Tran et al., 2019; Vidal-Sanz et al., 2017*). Following a crush injury, axotomized retinal ganglion cells (RGCs) face both intrinsic and environmental barriers to axon regeneration, and the pro-apoptotic aspects of this response ultimately result in extensive RGC neurodegeneration over subsequent weeks (*Syc-Mazurek et al., 2017a; Watkins et al., 2013; Welsbie et al., 2013; Welsbie et al., 2017*). Though neuronal knockout of DLK, c-Jun, or ATF3 each confers considerable RGC neuroprotection, disrupting these also limits the upregulation of regeneration-associated genes (RAGs) and the success of regenerative interventions, such as knockout of the tumor suppressor PTEN, that offer promise for enabling CNS repair (*Jacobi et al., 2022; Watkins et al., 2013*). Devising strategies to suppress neuronal loss without restricting intrinsic programs for repair will therefore require improving our understanding of which pathways and effectors within the injury response control axon regenerative and neurodegenerative programs and how these overlap (*Patel et al., 2020*).

Of particular interest is deciphering the contributions of the Integrated Stress Response (ISR). The ISR, activated in parallel to c-Jun as part of the DLK signaling cascade (*Larhammar et al., 2017*), results in the suppression of general translation but enhanced transcription and translation of select mRNAs, including *Atf4*, encoding the activating transcription factor-4 (ATF4), and *Ddit3*, encoding the C/EBP homologous protein (CHOP) (*Pitale et al., 2017*). Both ATF4 and CHOP have the potential to regulate transcription of ATF3 and/or heterodimerize with c-Jun, ATF3, other transcription factors of the C/EBP family, or each other, raising the possibility of crosstalk between the ISR and MAPK arms of the DLK response (*Pakos-Zebrucka et al., 2016*). Genetic disruption of ISR activation or of neuronal *Eif2ak3*, the gene encoding the ISR-activating kinase PERK, confers significant but incomplete RGC neuroprotection after optic nerve crush (*Larhammar et al., 2017; Yang et al., 2016*). These findings demonstrate that the neuronal ISR represents a functional component of the intrinsic neurodegenerative response but have provided limited mechanistic insight.

Recent studies emphasize the importance of defining the downstream effectors of the ISR, their relationships to c-Jun, and their impacts on RGC axon regenerative potential. Based on CRISPR screening, RNA-seq, and ATAC-seq, *Tian et al., 2022* proposed that ATF4 serves, along with C/EBPγ, as a critical component of a core neurodegenerative program, with CHOP and ATF3 together mediating a second core program (*Tian et al., 2022*). Injured RGCs expressing *Cas9* and a gRNA pool targeting ATF4 exhibited improved survival and modulation of genes primarily associated with intrinsic neuronal stressors (e.g. DNA damage response, autophagy, and the NAD/p53 pathway) distinct from canonical ATF4 target genes for adapting to amino acid deprivation, endoplasmic reticulum stress, and other ISR-activating insults. In addition to this, partial neuroprotection could also be achieved by gRNA pools targeting either Ddit3/CHOP or ATF3, either of which resulted in extensive, highly overlapping, transcriptional consequences related to cytokines and innate immunity (*Tian et al., 2024; Tian et al., 2022*). Importantly, a companion report shows that, of these same gRNA pools, only targeting ATF3 reduced RGC axon regeneration enabled by knockout of the tumor suppressor PTEN (*Jacobi et al., 2022*). Conditional knockout of ATF4 and CHOP has been reported to provide more neuroprotection than either alone, despite knockout of either appearing in that study to be sufficient to deplete the mRNA for both (*Fang et al., 2023*). Together, these findings led to the proposal that ATF4 and CHOP function in parallel neurodegenerative programs and, unlike ATF3 and c-Jun, do so without influencing RGC axon regenerative potential.

Other reports, however, seem to be inconsistent with these non-canonical roles for ATF4 and CHOP. The DLK-mediated stress response includes ISR-dependent upregulation of *Chac1*, *Ddit3*, and other target genes consistent with a typical ATF4-mediated transcriptional program, inclusive of CHOP activation (*Larhammar et al., 2017*). In addition, the proposed role of CHOP as a principal neuron-intrinsic effector of the response to optic nerve injury appears to be incongruent with the mild impact of germline *Ddit3*/CHOP knockout on that response. RNA-seq of retinae from these mice uncovered

few injury-induced transcripts that were significantly suppressed relative to the extensive impact of neural knockout of c-Jun (*Syc-Mazurek et al., 2022*). Though these CHOP-null mice reproducibly exhibit partial RGC neuroprotection (*Hu et al., 2012*; *Syc-Mazurek et al., 2017b*), the modest influence of CHOP deficiency on the transcriptome suggests the need for further evaluation of whether its primary role in regulating RGC survival is neuron-autonomous, as it may also reflect developmental effects or influences on other cell types, perhaps outside of the retina.

Here, we use previously validated conditional knockout (cKO) mice lines to directly address these and other questions raised by germline knockout and CRISPR studies. We find that a canonical ATF4 response functions as the principal mediator of a PERK-activated response that broadly contributes to both RGC apoptosis and axon regenerative potential. Unexpectedly, neuronal CHOP, long considered to be a primary neuron-intrinsic effector of the ISR in injured RGCs, plays a relatively modest role within these PERK/ATF4-stimulated programs. These findings reveal an integral role for PERK-stimulated ATF4 in the response to optic axon damage, serving to link neurodegenerative and axon regenerative transcriptional programs that determine the fates of distressed neurons.

## Results

### Neuronal ATF4 knockout mimics the neuroprotection provided by PERK deletion

Deletion of neuronal PERK, a key mediator of the ISR after optic nerve crush, provides significant, though incomplete, neuroprotection to RGCs (*Larhammar et al., 2017*). To determine how the ISR effectors ATF4 and CHOP contribute to neurodegeneration, we evaluated the survival of RGCs in cKO mice using immunohistochemistry for the pan-RGC marker RBPMS. We used high-titer intravitreal injection of AAV serotype 2 (AAV2) harboring the human synapsin-1 promoter (*hSyn1*), resulting in widespread Cre activity throughout neurons of the ganglion cell layer and inner nuclear layer of the retina (*Figure 1—figure supplements 1 and 2*). We confirmed the expected loss of *Atf4* transcript in ATF4 cKO mice and *Ddit3*/CHOP transcript in CHOP cKO mice by RNAScope (*Figure 1—figure supplements 3 and 4*). Evaluation of neuronal survival two weeks after optic nerve crush revealed that targeting ATF4, but not CHOP, confers significant neuroprotection of RBPMS-expressing RGCs (*Figure 1A*). We next compared the neuroprotection measured by this method with that determined using phospho-c-Jun as a robust, easily quantifiable nuclear marker of injured RGCs, finding similar improvements to RGC survival upon ATF4 knockout using either method (*Figure 1B*). To determine if CHOP disruption might augment the neuroprotection conferred by ATF4 disruption, we generated ATF4/CHOP double conditional knockout (dcKO) mice. However, these mice did not exhibit significantly greater RGC survival than ATF4 cKO mice (*Figure 1C*). Finally, we directly compared the effects of PERK knockout and ATF4 knockout, finding disruption of ATF4 is sufficient to afford a similar degree of neuroprotection to that of PERK (*Figure 1D*). Together with previous findings that ATF4 activation is DLK- and PERK-dependent (*Larhammar et al., 2017*), these results suggest that ATF4 serves as the primary mediator of the pro-apoptotic effects of the ISR after optic nerve crush.

### Neuronal PERK mediates an extensive contribution to the injury response primarily through ATF4

The equivalent neuroprotection afforded by knockout of PERK and ATF4 raises the possibility that ATF4 may be the principal neuron-autonomous mediator of ISR-activated transcription influencing RGC survival. To better understand the contributions of PERK and its effectors to the transcriptional response, we performed expression profiling of retina by RNA-seq three days after optic nerve crush in wild-type mice and three cKO lines: *Eif2ak3* cKO to disrupt PERK, *Ddit3* cKO to disrupt CHOP, and *Atf4* cKO to disrupt ATF4 (*Supplementary file 1*).

We began by determining the contribution of PERK and its effectors to the transcriptional injury response. Using a stringent statistical assessment for differentially expressed genes (DEGs, false discovery rate FDR <0.05), we identified 282 transcripts modulated by injury in control wild-type mice injected with AAV2-*hSyn1*-Cre. Among these, 117, or 41.5%, reached this same strict threshold for differential expression in a comparison between PERK cKO retinae after injury and wild-type retinae after injury (*Supplementary file 1*), with this number increased to 157, or 55.7%, using a moderately relaxed threshold of $p<0.01$ and FDR <0.2 (*Figure 2A*). All but three of these changed in the opposite

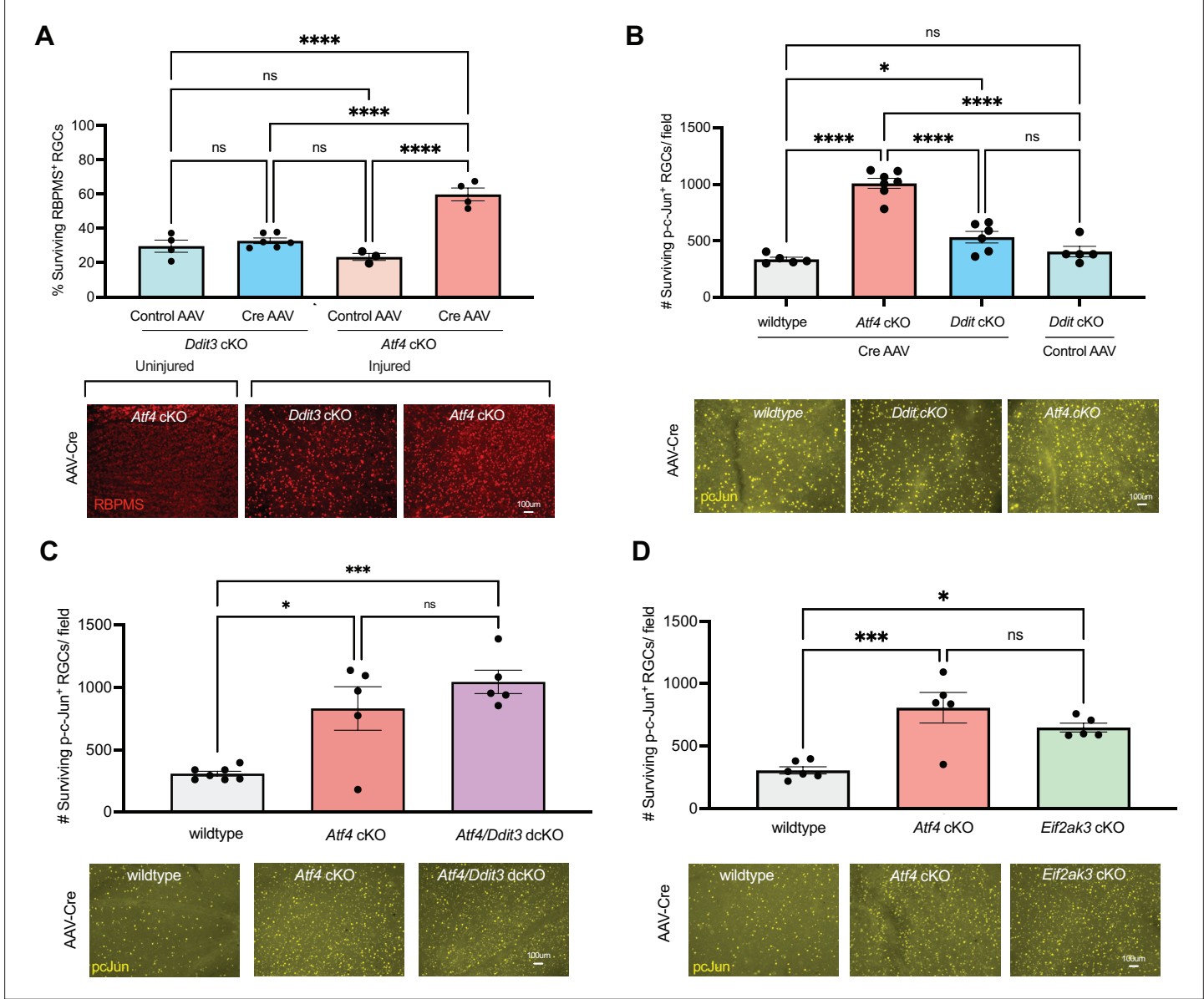

**Figure 1.** Neuronal activating transcription factor-4 (ATF4) contributes to retinal ganglion cell (RGC) neurodegeneration after optic nerve crush. (**A**) Neuronal knockout of ATF4, but not CHOP (*Ddit3*), by intravitreal AAV2-*hSyn1*-mTagBFP2-ires-Cre in conditional knockout (cKO) mice improves survival of RGCs 14 days after optic nerve crush, as assessed by immunostaining for the pan-RGC marker RBPMS (*n*=3-4 mice/condition). (**B**) Neuronal knockout of ATF4 similarly improves survival of RGCs when assessed by the robust nuclear marker of injured RGCs, phospho-c-Jun (p-c-Jun) (*n*=5-7 mice/condition). (**C, D**) Comparable levels of neuroprotection are conferred by ATF4 cKO and (**C**) ATF4/CHOP double cKO (dcKO), or (**D**) *Eif2ak3* cKO, which results in deletion of the Integrated Stress Response (ISR)-activating kinase PERK (*n*=5-7 mice/condition). Error bars are SEM. Statistical analysis: one-way ANOVA with Tukey's multiple comparison with a single pooled variance (*$p<0.05$; ***$p<0.001$; ****$p<0.0001$).

The online version of this article includes the following source data and figure supplement(s) for figure 1:

**Source data 1.** Four worksheets, one per figure panel, reporting calculated RGC survival values plotted in *Figure 1*.

**Figure supplement 1.** Intravitreal AAV2-*hSyn1*-Cre results in recombination of floxed alleles in the inner retina.

**Figure supplement 2.** Intravitreal AAV2-*hSyn1*-Cre results in recombination of floxed alleles in the inner retina.

**Figure supplement 3.** Intravitreal AAV2-*hSyn1*-Cre results in loss of *Atf4* mRNA in the retinal ganglion cell layer of activating transcription factor-4 (ATF4) conditional knockout mice.

**Figure supplement 4.** Intravitreal AAV2-*hSyn1*-Cre results in loss of *Ddit3*/CHOP mRNA in the retinal ganglion cell layer of C/EBP homologous protein (CHOP) conditional knockout mice.

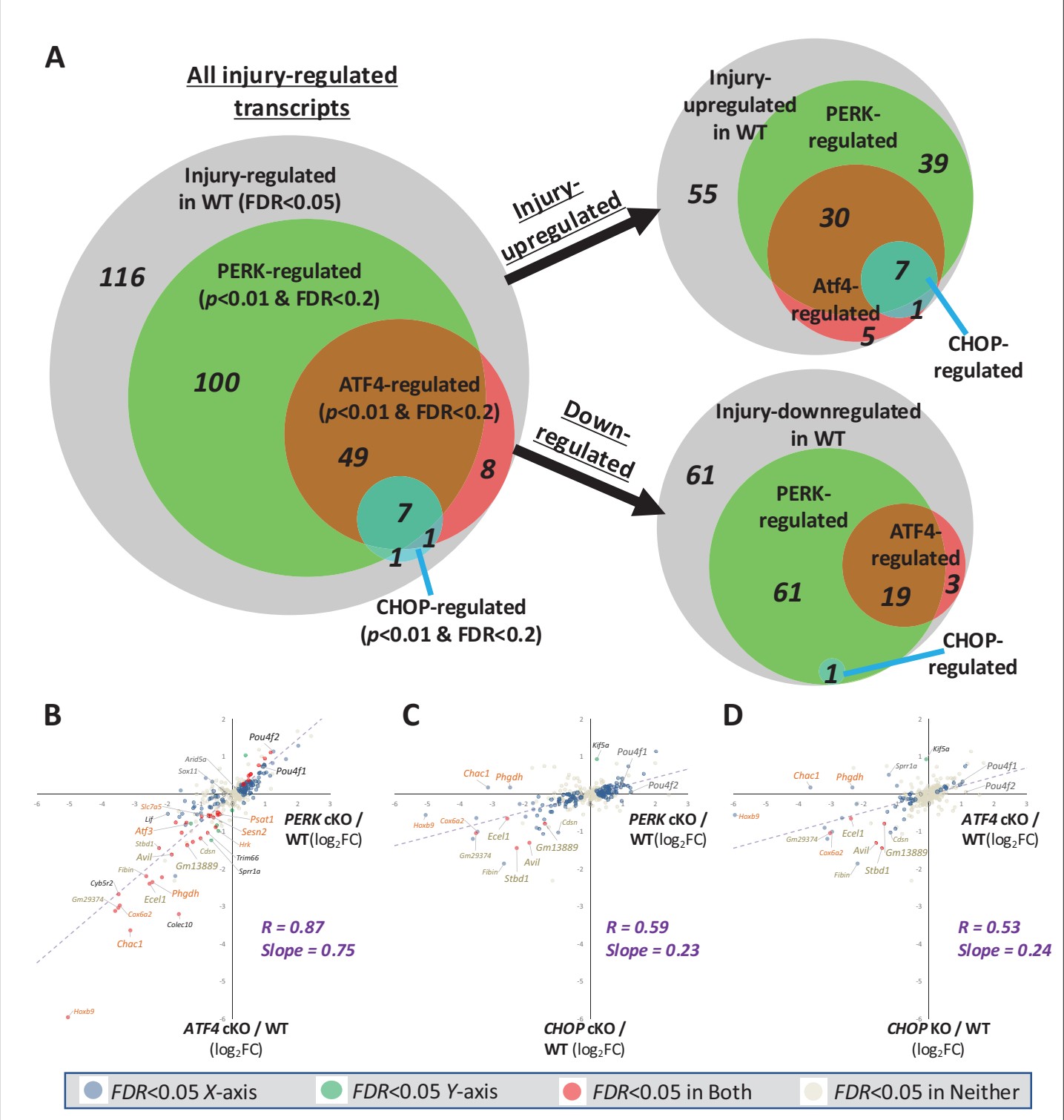

**Figure 2.** The PERK-activated Integrated Stress Response (ISR) is a prominent contributor to the transcriptional response to optic nerve injury, primarily acting through activating transcription factor-4 (ATF4). (**A**) Venn diagrams of 282 transcripts exhibiting significant differences by RNA-seq between uninjured retinae and 3 days after optic nerve crush (FDR <0.05). Amongst those, transcripts that exhibit differences between wild-type and conditional knockout (cKO) retinae after injury (p<0.01 and FDR <0.2) were classified as PERK-, ATF4-, or CHOP-regulated (n=4-5 retinae/condition). (**B–D**) Linear regression analyses comparing the impact of neuronal PERK deletion (*Eif2ak3* cKO), ATF4 deletion (*Atf4* cKO), and CHOP deletion (*Ddit3* cKO) on the same 282 injury-responsive transcripts (n=4-5 retinae/condition). Names of representative ATF4-dependent transcripts are burnt orange text and representative CHOP-dependent transcripts are gold text. Significant changes are indicated by colored dots (FDR <0.05).

direction of their modulation by injury, suggesting a central role for the ISR in mediating these injury responses. Among the 154 transcripts regulated by injury in a PERK-dependent manner, 56 (36.4%) are also suppressed by ATF4 knockout, with a particularly pronounced contribution among those transcripts that are upregulated after injury (37 out of 73, or 50.7%), including seven of the nine transcripts that display CHOP-dependence at a threshold of $p<0.01$ and FDR <0.2 (*Figure 2A*). Notably, among ATF4-regulated transcripts, 86.2% also exhibit PERK dependence at this threshold, consistent with ATF4 activation after optic nerve crush being mediated predominantly by PERK and not subject to compensation by other eIF2α kinases (*Larhammar et al., 2017*). This analysis, though limited by its arbitrary thresholds for defining factor-dependence, implies an essential role of the PERK-activated ISR in the transcriptional injury response, with CHOP serving as a relatively minor contributor within a program primarily mediated by ATF4.

To evaluate these relationships in a threshold-independent manner, we performed linear regression of the 282 injury-regulated genes. We first compared the impact of PERK cKO on the regulation of these transcripts by injury with that of ATF4 cKO. This analysis reveals a strong correlation ($R=0.87$) that, along with a best-fit slope of 0.75, suggests that the role of ATF4 in mediating the effects of PERK is more comprehensive than suggested by threshold-dependent analyses alone (*Figure 2B*). A similar assessment uncovers a milder ($R=0.59$), but still highly significant, correlation between PERK-dependent and CHOP-dependent injury-responsive transcripts (*Figure 2C*). However, the shallow slope of 0.23 for the best-fit line reveals that, though CHOP influences similar genes as PERK, it does so relatively mildly. Consistent with these findings, we observe a significant correlation between ATF4-dependent and CHOP-dependent transcripts, again with a shallow slope that indicates a more potent influence of ATF4 knockout (*Figure 2D*). These results imply that CHOP serves as a potentiator of the ISR after optic nerve crush rather than a primary mediator. Together, these findings are consistent with ATF4 exerting the dominant role in the transcriptional response and promotion of RGC apoptosis.

## PERK signaling regulates canonical ATF4 and c-Jun transcriptional programs

To explore the transcriptional programs downstream of PERK, we next applied the Upstream Regulator tool of Ingenuity Pathway Analysis (IPA), which includes determination of potential transcription factors likely to be mediators of the observed patterns of transcriptional changes and known target genes derived from extensive curation of the literature. Consistent with previous studies (*Syc-Mazurek et al., 2022*; *Yasuda et al., 2016*), we find that ATF4, along with c-Jun, emerges among the top five candidates for a positive transcriptional regulator ($p<10^{-13}$ and Activation Z-score >2.5), based on the upregulation of many genes previously demonstrated to be direct targets of ATF4, including *Atf3*, *Mthfd2*, *Stc2*, and *Phgdh* (*Ben-Sahra et al., 2016*; *Han et al., 2013*; *Pan et al., 2007*; *Zhao et al., 2016*; *Figure 3A*). IPA also suggested PGC1α (gene name: *Ppargc1a*), which can be negatively regulated by ATF4 activation (*Montori-Grau et al., 2022*; *Wang et al., 2013*), as the most influential negative transcriptional regulator (*Supplementary file 2*). CHOP (gene name: *Ddit3*) was implicated among a cluster of dozens of less influential positive regulators ($p<10^{-5}$ and Activation Z-score >1.25) (*Figure 3A*; *Supplementary file 2*). Comparisons between wild-type and PERK (*Eif2ak3*) cKO retinae 3 days after injury suggest that disruption of PERK abrogates ATF4 activation most strongly, with c-Jun and PGC1α also affected (*Figure 3B*, *Supplementary file 3*). This analysis suggests the activation of a canonical PERK/ATF4-mediated transcriptional response (*Figure 3C*) after RGC axon injury and raised the unexpected possibility that PERK signaling might also influence c-Jun-regulated target genes.

To further investigate a potential interaction between c-Jun- and PERK-regulated transcripts, we leveraged RNA-seq datasets from a separate, similarly designed, study that included mice lacking c-Jun in the majority of neural retina and CHOP-null mice (*Syc-Mazurek et al., 2022*). First, we evaluated the expression profiles of the wild-type injury conditions between these independent studies, finding an exceptional correlation ($R=0.95$, Slope = 1.07) among 282 injury-regulated genes (*Figure 3D*). That robust relationship between the control conditions provided a basis for further comparisons. We therefore proceeded with a cross-study analysis comparing the impact of PERK cKO with that of c-Jun cKO. That analysis revealed a significant correlation ($R=0.49$; Slope = 0.44) among c-Jun- and PERK-modulated transcripts (*Figure 3E*). Nevertheless, some injury-modulated transcripts are regulated exclusively by c-Jun (e.g. *Sox11* and *Arid5a*) or PERK (e.g. *Chac1* and *Phgdh*). Together, these findings suggest that, despite a lack of influence of c-Jun knockout on PERK (*Eif2ak3*), ATF4

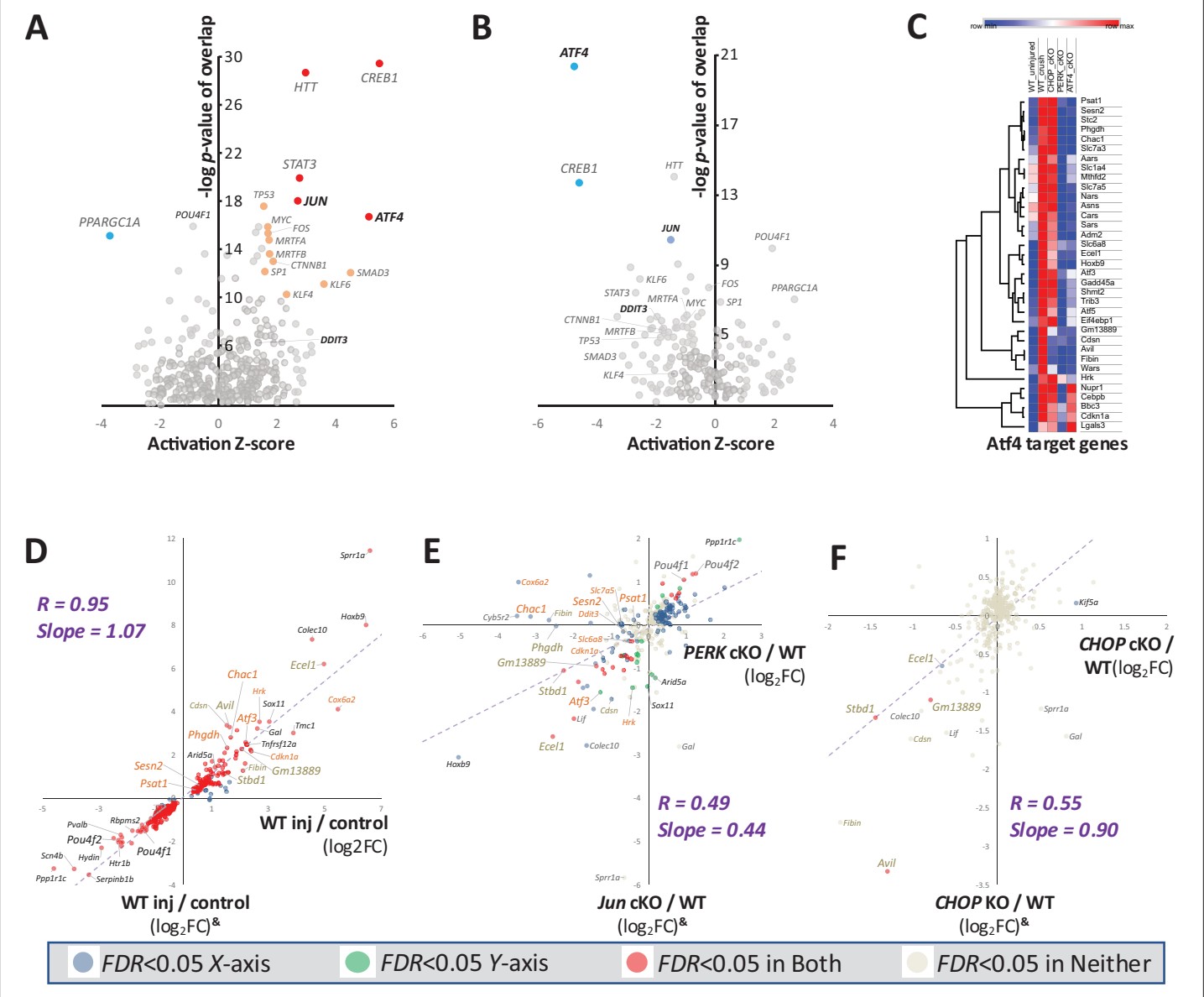

**Figure 3.** PERK regulates transcriptional changes through canonical activating transcription factor-4 (ATF4) target genes and influence on c-Jun-regulated programs. (**A, B**) Volcano plots of Ingenuity Pathway Analysis (IPA) implicate ATF4 and c-Jun as potential Upstream Transcriptional Regulators of injury-induced expression changes in (**A**) wild-type retina, and (**B**) those whose activity after injury is reduced by neuronal PERK deletion. Red, orange, blue, and light blue dots indicate Z>2.5, $p<10^{-13}$; Z>1.5, $p<10^{-10}$; Z<−2.5, $p<10^{-13}$; and Z<−1.5, $p<10^{-10}$, respectively. (**C**) Heat map of RNA-seq data, showing known and putative ATF4 target genes after injury in wild-type or conditional knockout (cKO) retinae. (**D–F**) Linear regression analyses comparing current whole retina RNA-seq cKO data to published data from a similarly designed experiment (*Syc-Mazurek et al., 2022*). Data from that independent study is indicated on the y-axis by &. (**D**) Strong correlation among 282 injury-responsive transcripts in the respective wild-type injury conditions between these two independent studies. Names of representative ATF4-dependent transcripts are burnt orange text and representative C/EBP homologous protein (CHOP)-dependent transcripts are gold text. (**E**) Significant correlations between the impacts of neuronal PERK conditional knockout (this study) and deletion of c-Jun from the majority of neural retina& ($p<0.001$). (**F**) Cross-study comparison of CHOP (*Ddit3*) cKO (this study) and CHOP (*Ddit3*) KO& shows strong correlation between the few CHOP-dependent injury-responsive transcripts (red dots).

(*Atf4*), or CHOP (*Ddit3*) transcripts and vice versa, these parallel pathways participate in substantially overlapping transcriptional programs.

We next evaluated the similarity between germline and neuronal knockout of CHOP (*Ddit3*) between these two studies. Though each uncovered only a small number of CHOP-dependent, injury-responsive transcripts, those few exhibit considerable overlap. Just four transcripts passed a stringent test for CHOP-dependence in retinae of CHOP/*Ddit3*[−/−] mice (*Syc-Mazurek et al., 2022*), three of

which – *Stbd1*, *Gm13889*, and *Avil* (also known as Doc6 for 'dependent on CHOP-6') – have been demonstrated to be upregulated neuron-autonomously by injury within RGCs by scRNA-seq (*Tran et al., 2019*). Using similar stringency (FDR <0.05), these three were among only five genes that we independently detected as injury-upregulated in a neuronally CHOP-dependent manner using CHOP cKO mice (*Figure 3F*). Additional injury-induced transcripts suppressed by greater than 50% in both CHOP cKO and CHOP KO (though without surpassing the FDR <0.05 threshold) include *Fibin* (*p*<0.0005 in both studies) and *Cdsn* (*p*<0.001 in both studies), genes for which ChIP-seq has previously identified CHOP binding near their transcriptional start sites under stress conditions (*Han et al., 2013*). These results show that, despite differences in neuroprotection between CHOP-null mice (*Hu et al., 2012*; *Syc-Mazurek et al., 2017b*) and neuronal targeting of CHOP in cKO retinae, their transcriptional consequences within the retina closely resemble one another.

Together, these expression profiles suggest that the PERK-mediated ISR is an integral contributor to the retinal transcriptional response after optic nerve injury, acting primarily through canonical ATF4 targets and interacting with c-Jun-mediated transcriptional programs.

## RGC-autonomous ATF4- and CHOP-dependent transcriptional changes are prominently represented by whole retina transcriptomics

Our targeted expression of Cre recombinase in cKO mice argues for a central role for the neuronal ISR in the response to optic axon injury, with whole retina transcriptomics likely reporting both these neuron-autonomous effects and secondary consequences to other retinal cells. To determine the extent to which ISR-dependent transcription uncovered by retinal RNA-seq represents RGC-autonomous expression changes, we next leveraged an independent single-cell RNA-seq (scRNA-seq) data set that details the injury-induced expression changes within subtypes of RGCs at 2 and 4 days after optic nerve crush (*Tran et al., 2019*).

We began by assessing the whole retina expression of transcripts that have been demonstrated to be differentially expressed by RGCs at 2 or 4 days after optic nerve crush (pseudo-bulk scRNA-seq |Diff>0.3| and FDR <$10^{-20}$). Of 615 such transcripts, 597 are represented in wild-type whole retina 3 days after injury. We find that 290 of these transcripts exhibit differential expression (|$\log_2$FC|>0.25) in retina, with all but three modulated by injury in the same direction as reported by scRNA-seq (*Figure 4A*). Moreover, the majority of the 282 injury-responsive transcripts we have identified in whole retina exhibit concordance with changes previously found by scRNA-seq of RGCs (*Figure 4—figure supplement 1*). Importantly, many transcripts identified as PERK-, ATF4-, and/or CHOP-dependent in this and other studies are among those that are upregulated neuron-autonomously within multiple subtypes of RGCs following optic nerve crush (*Figure 4B*). Accordingly, RNAScope confirms that the ATF4-dependent transcripts *Chac1*, *Ecel1*, and *Atf3* are elevated in the GCL in conjunction with downregulation of the RGC marker *Rbpms*, with these changes suppressed in ATF4 cKO retinae (*Figure 4C and D*). Furthermore, RNAScope corroborates that the *Avil* and *Ecel1* transcripts are upregulated in subsets of RGCs in an ATF4- and CHOP-dependent manner (*Figure 4D and E*; *Figure 4—figure supplement 2*). Consistent with this, the relationships we detected between PERK-dependent transcripts and ATF4- or c-Jun-dependent transcripts in whole retina are maintained when re-assessed using DEGs confirmed to be injury-regulated within RGCs by scRNA-seq (*Figure 4—figure supplement 3*). These results imply that the injury-induced ATF4 transcriptional program that we detect by whole retina transcriptomics of cKO mice primarily reflects RGC-autonomous expression changes.

This canonical ATF4 program, inclusive of the contribution of CHOP, contrasts with a report of two non-canonical, largely non-overlapping, programs controlled by these two ISR transcription factors (*Tian et al., 2022*). We next investigated potential sources of this discordance. The proposed parallel neurodegenerative programs were deduced in part by RNA-seq of FACS-enriched injured RGCs expressing gRNAs targeting ATF3 or *Ddit3/*CHOP. We therefore began with an assessment of how the injury responses detected in whole retina in the current study compare with those reported for RGCs collected by FACS. Using only transcripts independently identified as RGC-autonomous expression changes by scRNA-seq (*Tran et al., 2019*) for this cross-study comparison, we find a highly significant correlation (*R*=0.77) between injury-regulated transcripts under control conditions (i.e. without gene targeting) between these independent studies (*Figure 4—figure supplement 4*). This suggests that these distinct approaches similarly report prominent RGC-autonomous expression changes, inclusive of numerous known direct targets of ATF4, with

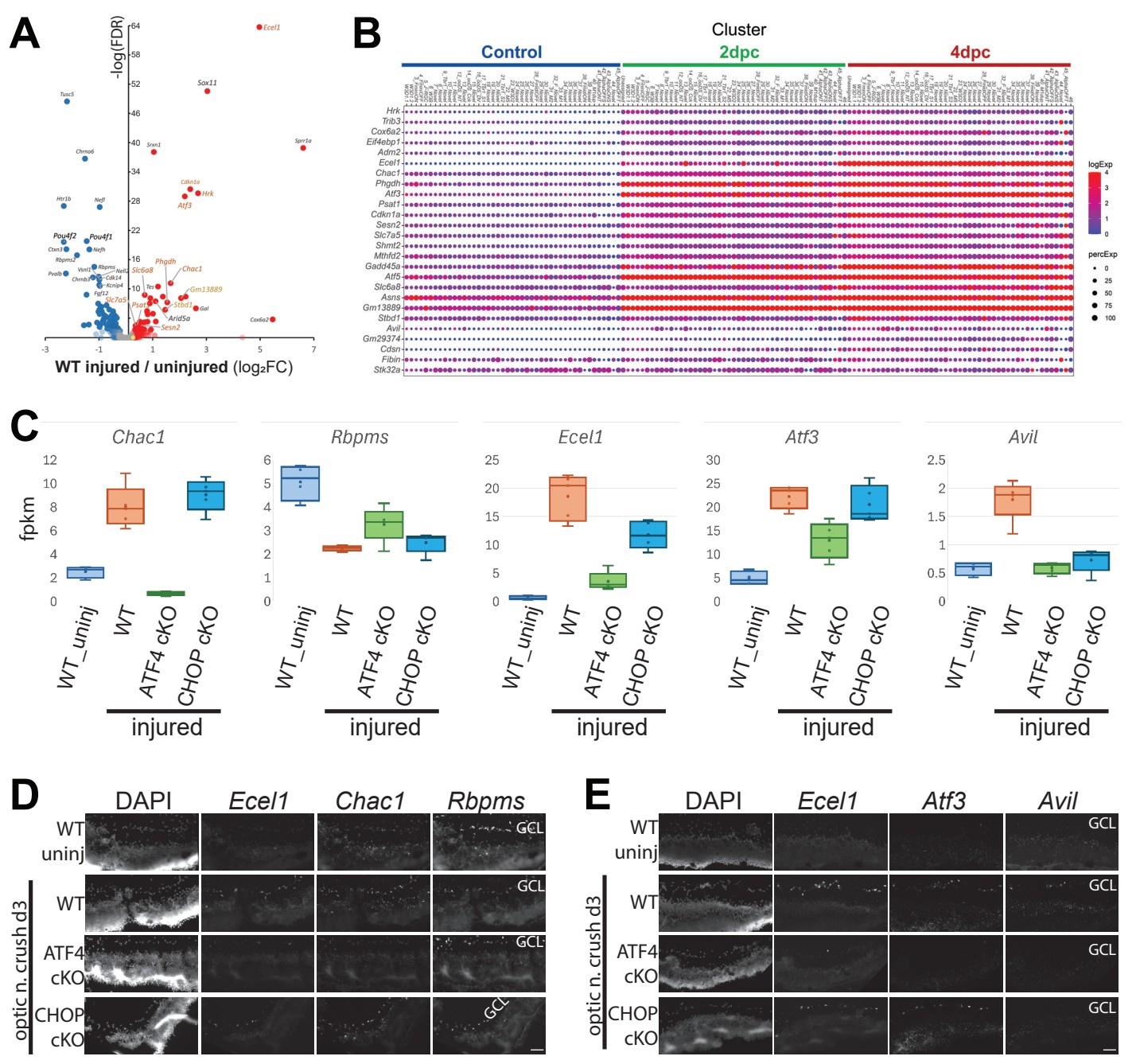

**Figure 4.** Cell-autonomous expression of activating transcription factor-4 (ATF4)- and C/EBP homologous protein (CHOP)-dependent transcripts by retinal ganglion cells (RGCs). (**A**) Volcano plot of retinal expression data three days after optic nerve crush (this study) for 597 transcripts that are significantly altered autonomously in RGCs at day 2 or day 4 after crush, as determined by single-cell RNA-seq (scRNA-seq) (*Tran et al., 2019*). 287 of these transcripts are detected at FDR <0.05 (red = upregulated, or blue = downregulated) or |log2FC|>0.25 (light red = upregulated, or light blue = downregulated) in whole retina. Only three transcripts (yellow dots) were significantly regulated in whole retina in the discordant direction from scRNA-seq findings. (**B**) scRNA-seq data set (*Tran et al., 2019*) reveals RGC-autonomous activation of canonical Integrated Stress Response (ISR) transcripts. Dot plot at 2 and 4 days after injury, showing both pan-RGC and type-specific upregulation of transcripts demonstrated by whole retina RNA-seq to be ATF4- and/or CHOP-dependent. Some transcripts exhibiting low, non-uniform, injury induction in RGCs – *Avil*, *Gm29374*, *Cdsn*, and *Fibin* – were amongst those that appear to exhibit potential CHOP-dependence but failed to reach threshold for inclusion in either retinal injury-dependence, CHOP-dependence, or scRNA-seq pseudo-bulk analyses. (**C–E**) Fluorescent in situ hybridization (RNAScope) of ATF4- and CHOP-dependent transcripts. (**C**) Box-and-whisker plots of whole retina transcriptomics (n=4-5 retinae per condition) for selected ATF4- and CHOP-dependent transcripts and the RGC marker gene *Rbpms* that were probed by RNAScope of fresh-frozen retinal cryosections. (**D, E**) Multiplex RNAScope across wild-type (WT) uninjured ('uninj') and 3 days post-crush ('3d') for three genotypes (WT, ATF4 cKO, and CHOP cKO) demonstrating prominent expression by *Rbpms⁺*

*Figure 4 continued on next page*

*Figure 4 continued*

RGCs of injury-induced transcripts that are reduced by knockout of ATF4 (*Atf3, Chac1*) or by knockout of either ATF4 or CHOP (*Ecel1, Avil*), concordant with RNA-seq findings. Nuclei are labeled with DAPI. Scale bars = 50 μm.

The online version of this article includes the following figure supplement(s) for figure 4:

**Figure supplement 1.** Prominent representation in retinal ganglion cells (RGCs) of injury-responsive transcripts detected by whole retina transcriptomics.

**Figure supplement 2.** Detection of *Avil* transcript by whole retina RNA-seq reflects injury-induced expression in a subset of retinal ganglion cells (RGCs).

**Figure supplement 3.** Linear regressions of PERK-, activating transcription factor-4 (ATF4)-, and c-Jun-dependent transcripts filtered by retinal ganglion cell (RGC)-autonomous injury-regulated transcripts.

**Figure supplement 4.** Strong concordance between this and an independent study regarding the retinal ganglion cell (RGC)-autonomous upregulation of numerous known activating transcription factor-4 (ATF4) target genes following optic nerve crush.

**Figure supplement 5.** Minimal concordance between this and an independent study regarding the ctivating transcription factor-4 (ATF4) dependence of known ATF4 target genes following optic nerve crush.

**Figure supplement 6.** Minimal concordance between this and an independent study regarding the C/EBP homologous protein (CHOP) dependence of retinal ganglion cell (RGC)-autonomous injury-induced changes following optic nerve crush.

a steep linear regression (slope = 2.0) that is consistent with greater sensitivity for these changes by enrichment for RGCs by FACS. Despite that similarity, linear regression reveals little correlation between the impact of cKO- and gRNA-mediated knockout on 597 RGC-autonomous, injured-regulated transcripts. We find only a very weak, shallow correlation between the effects of ATF4 gRNA and *ATF4* cKO ($R=0.25$, slope = 0.52). cKO-sensitive signature ATF4 target genes *Chac1*, *Phgdh*, *Sesn2*, *Slc7a5*, and *Asns*, amongst others, are not among the 300 of these 598 (50.2%) transcripts reported to be significantly affected by ATF4 gRNA (*Figure 4—figure supplement 5*; *Crawford et al., 2015*; *Garaeva et al., 2016*; *Han et al., 2013*; *Liu et al., 2018*; *Oh-Hashi et al., 2013*; *Park et al., 2017*). We also find no correlation between the influence of CHOP/*Ddit3* cKO and gRNA targeting CHOP, with no reported inhibition by this gRNA pool of the CHOP KO- and cKO-sensitive transcripts *Gm13889* and *Stbd1* among 333 (55.7%) significantly affected transcripts (FDR <0.05) (*Figure 4—figure supplement 6*). Notably, the reported effects of gRNA targeting CHOP, like ATF3 gRNA, include minimal reduction of *Ddit3*/CHOP transcript but greater than 70% suppression of the pro-apoptotic and pro-regenerative *Atf3* transcript (*Tian et al., 2022*), an effect not observed in CHOP/*Ddit3* germline (*Syc-Mazurek et al., 2022*) or conditional knockout mice (present study), despite ample sensitivity for *Atf3* mRNA modulation using whole retina transcriptomics. These analyses argue that more specific and effective disruption of target genes by cKO may contribute to the striking discordance between the transcriptional programs of ATF4 and CHOP suggested by these distinct approaches.

## Neuronal ATF4 knockout limits axon regeneration by PTEN-deficient RGCs

The commonalities that we uncovered between c-Jun- and PERK-regulated transcription suggested the hypothesis that, like c-Jun, PERK/ATF4 could be involved not only in promoting apoptosis but also in promoting regeneration. To investigate this possibility, we first examined the PERK- and ATF4-dependence of regeneration-associated genes (RAGs) and the injury-induced downregulation of mature and subtype-specific RGC markers, finding that many exhibit at least partial PERK- and ATF4-dependence (*Figure 5A and B*). We therefore crossed mice harboring the floxed PTEN alleles and those harboring floxed ATF4 alleles or floxed PERK alleles to generate homozygous dcKO mice. Following intravitreal injection of AAV2-*hSyn1*-Cre, we performed optic nerve crush, comparing axon regeneration in these dcKO mice to that of *PTEN* cKO mice (*Figure 5C and D*). We find that neuronal PERK deficiency or ATF4 deficiency limits the efficacy of this regenerative intervention. These data support an integral contribution of the ISR to the axon regenerative program in these CNS neurons and indicate that, as with DLK and c-Jun, ATF4 is involved in both pro-regenerative and pro-apoptotic transcriptional changes.

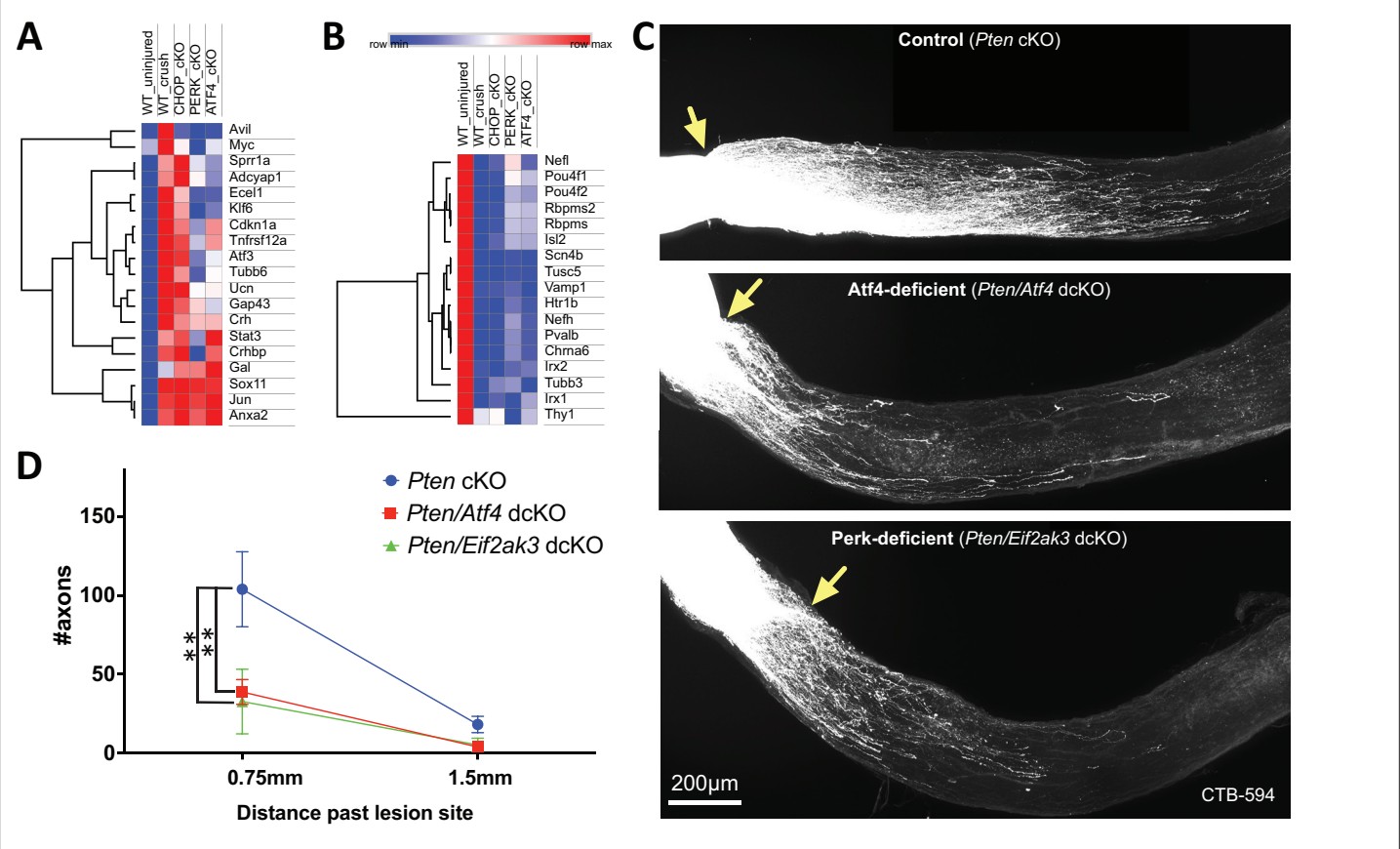

**Figure 5.** PERK-ATF4 contributes to retinal ganglion cell (RGC) axon regenerative potential after optic nerve crush. (**A, B**) Heat maps of select injury-responsive transcripts revealed by retinal RNA-seq three days after optic nerve crush in wild-type and cKO mice, focusing on (**A**) genes implicated in axon regeneration, and (**B**) the maintenance of mature and subtype-specific RGC phenotypes. (**C, D**) Double conditional knockout (dcKO) of neuronal PERK (*Eif2ak3*; n=4) or ATF4 (*Atf4*; n=7) reduces RGC axon regeneration compared to conditional knockout (cKO) of the tumor suppressor PTEN alone (n=8). Error bars are SEM. Statistical analysis: two-way ANOVA with Sidak's multiple comparisons test (\*\*$p<0.01$).

The online version of this article includes the following source data for figure 5:

**Source data 1.** Measurements of axons growing 0.75-mm and 1.5-mm past the injury site 15 days after optic nerve crush, graphed in *Figure 5D*.

## Discussion

Cellular stress signaling pathways typically couple the promotion of growth and repair with priming for apoptosis, a feature of stress responses that may aid in the elimination of cells that are irretrievably damaged (*Hotamisligil and Davis, 2016*). As prominent effectors of stress signaling, transcription factors of the bZIP family, including c-Jun and ATF3, therefore, can contribute either to apoptosis or to recovery, depending on context, complicating efforts to harness transcriptional programs for axon regeneration independently of neurodegenerative programs (*Jacobi et al., 2022*; *Kole et al., 2020*; *Raivich et al., 2004*; *Simon and Watkins, 2018*; *Syc-Mazurek et al., 2017b*; ; *Tian et al., 2022*; *Watkins et al., 2013*). Multiple lines of evidence indicating that the PERK-activated ISR contributes to RGC apoptosis after optic nerve crush raised the appealing possibility that this pathway may represent a distinct branch of the injury response that might be targeted to selectively reduce neurodegeneration without suppressing repair programs (*Hu et al., 2012*; *Larhammar et al., 2017*; *Tian et al., 2022*; *Wang et al., 2020*). The cKO studies reported here, however, reveal that the PERK-activated ISR is more integral to the entire RGC injury response than previously appreciated, contributing substantially, in conjunction with c-Jun, to both apoptotic and regenerative programs. Though MAP kinase stress signaling and the ISR are known to exhibit some degree of crosstalk in other settings (*Brown et al., 2016*; *Danzi et al., 2018*; *Pakos-Zebrucka et al., 2016*), the exceptionally close coupling of these largely independent pathways by DLK signaling seems to be a critical feature of the response of

RGCs to axon injury, with disruption of ATF4, like that of c-Jun, limiting both neurodegeneration and axon regenerative potential.

By examining in parallel the neuronal knockout of PERK and its effectors ATF4 and CHOP, along with comparisons to related studies, the current work has provided insights into the mechanisms by which the ISR influences the fates of injured RGCs. These conditional knockout mouse experiments reveal that ATF4 is the principal effector of the ISR transcriptional response, with CHOP playing a modest role within a canonical ATF4-mediated program. Though RGC neuroprotection in germline CHOP knockout mice has long been interpreted as evidence of its central role in the neuronal ISR, our findings suggest that further investigations of CHOP may instead uncover its non-autonomous mechanisms regulating the survival of injured RGCs. The unexpectedly subtle role for neuronal CHOP aligns well with expression profiling data from CHOP-null mice (*Syc-Mazurek et al., 2022*) and its modest role in this study in determining RGC survival, but it appears to contrast with the more significant role for CHOP, additive with ATF4, suggested by other studies (*Fang et al., 2023*; *Tian et al., 2022*). One factor that may influence the results of CHOP cKO experiments is the approximately twofold elevated expression of a collection of eight genes (*B4galnt1*, *Slc26a10*, *Arhgef25*, *Dtx3*, *F420014N23Rik*, *Pip4k2c*, *Kif5a*, and *Dctn2*) near the *Ddit3/*CHOP locus on chromosome 10. One of those genes, *Kif5a*, is injury-responsive and alterations to its expression can influence RGC survival (*Shah et al., 2022*). This confound may be expected to have also been present in an independent study using the same mouse model and is therefore not likely to account for the reported differences in RGC neuroprotection (*Fang et al., 2023*). Though the apparent reduction of ATF3 reported for gRNA targeting *Ddit3/*CHOP, not seen in CHOP KO or cKO models, may suggest an explanation for its strikingly similar reported effects of ATF3 gRNA on the transcriptome and RGC survival, those results seem incongruent with the differential reported effects of ATF3 and CHOP gRNA on RGC axon regeneration (*Jacobi et al., 2022*; *Tian et al., 2022*). With cKO experiments revealing instead that ATF3 upregulation is dependent on ATF4 and c-Jun, it will be of interest to elucidate how much of the commonality among the transcriptional programs of these bZIP factors is attributable to interdependence of their expression and how much to their combinatorial heterodimerization. Among other non-exclusive possibilities, the commonalities between ATF4- and c-Jun-dependent transcription could reflect: (1) dependence of ATF3 elevation on c-Jun-ATF4 heterodimers; (2) dependence on ATF4 for induction of ATF3, which then heterodimerizes with c-Jun; or (3) regulation of common target genes through distinct DNA binding sites. The current study provides a framework for deciphering these networks in part by leveraging cross-study comparisons of retina, RGC, and single-cell RNA-seq data to uncover robust RGC-autonomous expression signatures of ATF4 and other transcription factors.

Given the potential for the ISR to influence neuronal fate through multiple translational and transcriptional mechanisms, it is perhaps surprising that its primary contributions to the fates of injured RGCs are mediated by a single transcription factor, ATF4. Nevertheless, at least four lines of evidence contend that the cKO experiments reported here provide a reliable picture of its contribution. First, the extensive commonality in injury-regulated transcripts across studies, whether utilizing retinal expression data or filtering for RGC-autonomous expression changes, argues against AAV2-*hSyn1*-Cre or other technical aspects of our approach altering the injury response. Second, support for the modest impact of CHOP disruption on RGC injury-regulated genes is provided by the highly similar findings in the independently generated and validated CHOP-null and CHOP cKO mouse lines. Third, we find remarkable overlap between expression changes and phenotypes mediated by PERK and ATF4, which we previously demonstrated to be in the same pathway after optic nerve crush and other DLK-activating insults (*Larhammar et al., 2017*). Finally, the interaction between PERK/ATF4 and c-Jun-regulated transcription uncovered by our cross-study analysis predicts an impact of PERK or ATF4 knockout on RGC axon regenerative potential that is validated by dcKO with PTEN. This result is consistent with recent findings that early injury responses, including regenerative and degenerative responses that we have found in this study to be driven by PERK/ATF4, are similar in non-regenerating wild-type and regenerating PTEN-deficient RGCs (*Jacobi et al., 2022*). Together, these conditional knockout studies highlight the critical roles for ISR-activated transcriptional programs in determining

the fates of distressed neurons and the intimate link between injury-activated neurodegenerative and axon regenerative programs.

# Materials and methods

## Key resources table

| Reagent type (species) or resource | Designation | Source or reference | Identifiers | Additional information |
|---|---|---|---|---|
| Genetic reagent (*Mus musculus*) | C57BL/6-Atf4tm1.1Cmad/J (Atf4 cKO /Atf4fl/fl) | Christopher Adams, University of Iowa (*Ebert et al., 2012*) | RRID:IMSR_ JAX:033380 | Also available Jackson Laboratory |
| Genetic reagent (*Mus musculus*) | B6.Cg-Ddit3tm1.1Irt/J (Chop cKO / Ddit3fl/fl) | Jackson Laboratory (*Zhou et al., 2015*) | RRID:IMSR_ JAX:030816 | |
| Genetic reagent (*Mus musculus*) | Eif2ak3tm1.2Drc/J (Perk cKO / Eif2ak3fl/fl) | Jackson Laboratory (*Zhang et al., 2002*) | RRID:IMSR_ JAX:023066 | |
| Genetic reagent (*Mus musculus*) | B6.129S4-Ptentm1Hwu/J (Pten cKO / Ptenfl/fl) | Jackson Laboratory (*Lesche et al., 2002*) | RRID:IMSR_ JAX:006440 | |
| Genetic reagent (*Mus musculus*) | B6.Cg-Gt(ROSA)26Sortm14(CAG-tdTomato)Hze/J (Ai14 Rosa26-LSL-tdtomato) | Jackson Laboratory (*Madisen et al., 2010*) | RRID:IMSR_ JAX:007914 | |
| Antibody | Phospho-c-Jun (Ser73) (D47G9) Rabbit Monoclonal #3270 | Cell Signaling Technology | RRID:AB_2129575 | 1:800 |
| Antibody | Anti-RBPMS Guinea Pig polyclonal #1832-RBPMS | Phospho Solutions | RRID:AB_2492226 | 1:250 |
| Antibody | Purified anti-Tubulin β3 (TUBB3) [TUJ1]; Mouse Monoclonal #801202 | Biolegend | RRID:AB_10063408 | 1:1250 |
| Antibody | NeuN (D4G4O) Rabbit Monoclonal #24307 | Cell Signaling Technology | RRID:AB_2651140 | 1:800 |
| Antibody | Alexa Fluor 647 Anti-BRN3A Rabbit Monoclonal [EPR23257-285] ab300744 | abcam | RRID:AB_2916038 | 1:100 |
| Sequence-based reagent | RNAScope probe Mm-Rbpms-C3 | Advanced Cell Diagnostics | Cat. No. 527231-C3 | |
| Sequence-based reagent | RNAScope probe Mm-Ecel1-C2 | Advanced Cell Diagnostics | Cat. No. 475331-C2 | |
| Sequence-based reagent | RNAScope probe Mm-Ecel1-C3 | Advanced Cell Diagnostics | Cat. No. 475331-C3 | |
| Sequence-based reagent | RNAScope probe Mm-Atf4 | Advanced Cell Diagnostics | Cat. No. 405101 | |
| Sequence-based reagent | RNAScope probe Mm-Chac1 | Advanced Cell Diagnostics | Cat. No. 514501 | |
| Sequence-based reagent | RNAScope probe Mm-Avil-C2 | Advanced Cell Diagnostics | Cat. No. 498531-C2 | |
| Sequence-based reagent | RNAScope probe Mm-Atf3 | Advanced Cell Diagnostics | Cat. No. 426891 | |
| Sequence-based reagent | RNAScope probe Mm-Ddit3 | Advanced Cell Diagnostics | Cat. No. 317661 | |

## Animals

All animals used were in C57BL6/J background strain. Animal care and experimental procedures were approved by the institutional animal care and use committee (IACUC) at Baylor College of Medicine, according to NIH Guidelines.

## Adeno-associated virus (AAV)

Adeno-associated viral vectors containing human synapsin-1 (*hSyn1*) promoter driving expression of mTagBFP2-IRES-Cre or mTagBFP2-IRES-NLS-smGFPmyc(dark) were packaged into AAV2 capsids at the Optogenetics and Viral Vector Core at Duncan Neurological Research Institute, Houston, Texas, USA.

## Intravitreal injections

Animals were anesthetized with isoflurane. Artificial tears ointment (Covetrus #11695-6832-1) was applied to the eyes. In the case of bilateral injections, the non-injected eye received artificial tears. The eyes were sterilized and prepared by three repeated applications of 5% ophthalmic betadine (Henry Schein #6900250), followed by Opti-clear ophthalmic eye wash (Akorn #NDC 17478-620-04) and wiped dry. Topical anesthetic 0.5% proparacaine HCl ophthalmic solution (Henry Schein #1365345) was applied. A 5 µl Hamilton syringe loaded with a custom 33-gauge needle (Hamilton #7803–5) was

used to puncture the eyeball and relieve some of the intraocular pressure. The Hamilton needle was inserted back into the same puncture site and 2 µl of AAV in titers ranging from $10^{12}$ to $10^{13}$ vg/ml was delivered per eye.

### Intra-orbital optic nerve crush (ONC)

Animals were dosed with 1 mg/kg buprenorphine sustained release formulation 1 hr prior to surgery. At the time of surgery, they were anesthetized with isoflurane. Artificial tears ointment (Covetrus #11695-6832-1) was applied to the non-surgical eye. The surgical eye was sterilized and prepared by three repeated applications of 5% ophthalmic betadine (Henry Schein #6900250), followed by Opti-clear ophthalmic eye wash and wiped dry. Topical anesthetic 0.5% proparacaine HCl ophthalmic solution (Henry Schein #1365345) was applied on the surgical eye (left). Incisions were made in both the conjunctival layers using a pair of Vannas scissors (World Precision Instruments #501777). Two pairs of suture-tying forceps (Fine Science Tools #1106307) were used to gently clear the soft tissue in the intra-orbital space behind the eye until the optic nerve was visible. A pair of Dumont forceps (Fine Science Tools #1125325) were used to manually crush the optic nerve for 5 s. The eyeball was gently pushed back into the orbit. Animals were then returned to their home cages and monitored until sternal recumbency was observed.

### Immunolabeling of retinae

Animals were euthanized by anesthesia overdose, followed by decapitation. The eyes were harvested and fixed in 4% paraformaldehyde for 1 hr. The retinae were dissected out in 1 X phosphate-buffered saline (PBS) and then blocked for 30 min in 5% goat serum, 0.5% Triton X-100, and 0.025% sodium azide in 1X PBS. The retinae were then incubated for 5 days in primary antibody diluted in blocking buffer solution at 4°C. They were then washed three times in 1X PBS with 0.5% Triton X-100 for 30 min each, and then moved to appropriate secondary antibody solution prepared in blocking buffer and incubated overnight. This was then followed by three washes again with 1X PBS with 0.5% Triton X-100 for 30 min each. Retinae were then mounted in Drop-n-Stain EverBrite Mounting Medium (Biotium #23008) onto slides and imaged using a Zeiss Axio Imager Z1 fluorescence microscope.

### Fluorescent in situ hybridization

Cryosections (12 µm) of fresh-frozen control retinae and retinae 3 days after optic nerve crush were subjected to RNAScope fluorescent in situ hybridization using the RNAScope Multiplex Fluorescent Reagent Kit V2 (Advanced Cell Diagnostics, Hayward, CA, USA) according to the manufacturer's protocols. The following probes were purchased from Advanced Cell Diagnostics: Mm-Rbpms-C3 (Cat No. 527231-C3), Mm-Ecel1-C2 (Cat No. 475331-C2), Mm-Ecel1-C3 (Cat No. 475331-C3), Mm-Atf4 (Cat No. 405101), Mm-Chac1 (Cat No. 514501), Mm-Avil-C2 (Cat No. 498531-C2), Mm-Atf3 (Cat No. 426891), and Mm-Ddit3 (Cat No. 317661).

### RGC survival assessment

Animals that underwent optic nerve crush surgeries were euthanized two weeks post-crush by anesthesia overdose, followed by decapitation. Retinae were dissected out, fixed, and immunolabeled as described above. Whole-mounted retinae were imaged uniformly from the periphery to the center, with 3–4 images captured at 10 x per leaflet of the flat-mounted retina (7–11 images per retina) in all fluorescently labeled channels. Images were blind coded and quantified for surviving RBPMS-positive and/or p-c-Jun-positive cells through a combination of hand counting and RGC counter tool of the Simple RGC plug-in in ImageJ FIJI. The average count of all images per retina was calculated to obtain mean cell count per image and plotted to show RGC survival. For the plot showing percentage survival of RGCs, the counts were normalized to mean count of uninjured retinae for each experimental group. Plots and statistical analyses were generated using GraphPad Prism.

### Axon regeneration assessment

Animals that underwent optic nerve crush surgeries were intravitreally injected on the surgical eye with 2 µl of Alexa 594-conjugated cholera toxin β (Life Technologies #C22842) at day 14. Animals were euthanized on day 15 by anesthesia overdose followed by decapitation. The surgical eye and optic nerve were dissected out together by first cutting the optic nerves at the chiasm through intracranial

dissection and making a single cut along the length of the control uninjured optic nerve. Incisions were then made in the orbital bones to remove the ceiling of the orbital cavity and gently release the surgical eye and optic nerve out together. The retinae and optic nerves were then drop-fixed together in 4% PFA. The connective tissue around the optic nerves were dissected out. The optic nerves were then processed for tissue clearing by incubating in 100% methanol for 4 min and then moved to Visikol-1 overnight on a shaker at 4°C. They were then moved to Visikol-2 for 2 hr with shaking at room temperature and then mounted in Visikol-2 onto slides. The whole-mounted and cleared optic nerves were imaged using a Zeiss Imager Z2 fluorescence microscope and Apotome 2.0. The nerves were imaged by capturing 3–5 sequential Z-stacks at 10x magnification along the length of nerve with a range of 200 µm and Z-stacks of 100 images each. The Z-stacks were stitched together using the FIJI stitching algorithm, and max intensity projections were created in FIJI. The optic nerve crush site was identified by the point where the brightest CTB labeling ends. The line tool in FIJI was used to mark regions 0.75 mm and 1.5 mm from the crush site. A vertical line was drawn across the optic nerve at these respective distances, and the number of axons that coursed through the drawn line at each distance was manually counted. In regions where there were too many axons that could be accurately counted, the Z-stacks were used for quantification. Every tenth images in the z-stack was counted for number of axons coursing through the drawn lines at 0.75 mm and 1.5 mm from the crush site, and the total number from each Z-stack was calculated to provide an estimated number of axons. Plots and statistical analyses were generated using GraphPad Prism.

## RNA preparation and Next-generation sequencing

Animals that underwent optic nerve crush surgeries were euthanized 3 days post-crush by anesthesia overdose followed by decapitation. The retinae were quickly dissected out in 1X PBS and snap frozen in dry ice-ethanol bath and stored in –80°C until ready to be processed for RNA extraction. RNA was extracted using the Qiagen RNeasy micro kit (#74004). Extracted RNA was sent to Novogene Co., Ltd (Beijing, China) for Next Generation Sequencing using their Illumina NovaSeq platform for mouse mRNA sequencing. RNA libraries were prepared according to Novogene procedures by polyA capture (or rRNA removal) and reverse transcription of cDNA. Illumina PE150 technology was employed to sequence the samples. Sample reads were aligned to mouse reference genome using the HISAT2 algorithm. Gene expression analysis was performed using the Novogene pipeline. Venn diagrams were generated using DeepVenn (*Hulsen, 2022*), heatmaps were generated using Morpheus (https:// software.broadinstitute.org/morpheus), pathway analyses were performed using Ingenuity pathway analysis (Qiagen) and single-cell mouse retinal ganglion cell atlas was obtained from the Broad Single Cell Portal (*Tarhan et al., 2023*), representing data from GSE137400 (*Tran et al., 2019*). Cross-study analyses of RNA-seq findings were performed using logFC and FDR values reported for c-Jun cKO and CHOP KO mice (*Syc-Mazurek et al., 2022*) and after performing an independent DESeq2 analysis of GSE190667 and GSE184547 FASTQ (*Tian et al., 2024*; *Tian et al., 2023*; *Tian et al., 2022*), uploading directly from GEO to the OneStopRNAseq resource (*Li et al., 2020*).

## Acknowledgements

We thank Katie Steck, Shivani Kulkarni, and Talia Sisroe for technical assistance. This work was supported by grants from Mission Connect, a project of the TIRR Foundation, NIH grants R01NS112691 and R01NS076708 to TAW, the Glaucoma Research Foundation, and NIH grant R00EY029360 to NMT. Research reported in this publication was supported by the Eunice Kennedy Shriver National Institute of Child Health & Human Development of the National Institutes of Health under Award Number P50HD103555 for use of the Neuroconnectivity Core facilities. The content is solely the responsibility of the authors and does not necessarily represent the official views of the National Institutes of Health.

## Additional information

### Funding

| Funder | Grant reference number | Author |
|---|---|---|
| Mission Connect, a project of the TIRR Foundation | | Trent A Watkins |
| National Institutes of Health | R01NS112691 | Trent A Watkins |
| National Institutes of Health | R01NS076708 | Trent A Watkins |
| Glaucoma Research Foundation | | Trent A Watkins |
| National Institutes of Health | R00EY029360 | Nicholas M Tran |

The funders had no role in study design, data collection and interpretation, or the decision to submit the work for publication.

### Author contributions

Preethi Somasundaram, Conceptualization, Formal analysis, Investigation, Methodology, Writing – original draft, Writing – review and editing; Madeline M Farley, Formal analysis, Investigation, Methodology, Writing – review and editing; Melissa A Rudy, Investigation, Visualization, Writing – review and editing; Katya Sigal, Shufang Wang, Investigation, Methodology; Andoni I Asencor, Methodology; David G Stefanoff, Investigation, Visualization; Malay Shah, Puneetha Goli, Jenny Heo, Investigation; Nicholas M Tran, Formal analysis, Visualization, Methodology, Writing – review and editing; Trent A Watkins, Conceptualization, Formal analysis, Supervision, Funding acquisition, Investigation, Visualization, Writing – original draft, Project administration

### Author ORCIDs

Preethi Somasundaram ![ORCID] https://orcid.org/0000-0002-2212-2071
Madeline M Farley ![ORCID] https://orcid.org/0000-0003-4852-2919
Melissa A Rudy ![ORCID] https://orcid.org/0000-0003-0165-5669
Katya Sigal ![ORCID] https://orcid.org/0009-0004-2953-0440
Andoni I Asencor ![ORCID] https://orcid.org/0000-0002-4835-0849
David G Stefanoff ![ORCID] https://orcid.org/0009-0008-4366-7645
Malay Shah ![ORCID] https://orcid.org/0000-0002-0284-3402
Puneetha Goli ![ORCID] https://orcid.org/0000-0001-5031-169X
Jenny Heo ![ORCID] https://orcid.org/0009-0000-9773-071X
Shufang Wang ![ORCID] https://orcid.org/0009-0003-2983-7453
Nicholas M Tran ![ORCID] https://orcid.org/0000-0002-4213-3499
Trent A Watkins ![ORCID] https://orcid.org/0000-0001-6723-3712

### Ethics

This study was performed in accordance with the Guide for the Care and Use of Laboratory Animals of the NIH, under approved IACUC protocol AN-7208 of Baylor College of Medicine. All surgery was performed under isoflurane anesthesia with appropriate use of analgesics.

Reviewer #1 (Public review): https://doi.org/10.7554/eLife.87528.3.sa1
Reviewer #2 (Public review): https://doi.org/10.7554/eLife.87528.3.sa2
Author response https://doi.org/10.7554/eLife.87528.3.sa3

## Additional files

### Supplementary files

Supplementary file 1. RNA-seq of whole retina 3 days after optic nerve crush across four different conditions: wild-type injured compared to wild-type uninjured; ATF4 cKO injured compared to

wild-type injured; PERK cKO injured compared to wild-type injured; CHOP cKO injured compared to wild-type injured.

Supplementary file 2. Results of the Upstream Regulator tool of Ingenuity Pathway Analysis (IPA) for transcripts significantly regulated by injury in wild-type retina 3 days after optic nerve crush.

Supplementary file 3. Results of the Upstream Regulator tool of Ingenuity Pathway Analysis (IPA) for differentially expressed transcripts between PERK cKO and wild-type retinas 3 days after optic nerve crush.

MDAR checklist

## Data availability

RNA-sequencing data is available from the Gene Expression Omnibus through series accession number GEO: GSE223321.

The following dataset was generated:

| Author(s) | Year | Dataset title | Dataset URL | Database and Identifier |
|---|---|---|---|---|
| Somasundaram P, Rudy MA, Tran NM, Watkins TA | 2023 | Coordinated stimulation of axon regenerative and neurodegenerative transcriptional programs by Atf4 following optic nerve injury | https://www.ncbi.nlm.nih.gov/geo/query/acc.cgi?acc=GSE223321 | NCBI Gene Expression Omnibus, GSE223321 |

The following previously published datasets were used:

| Author(s) | Year | Dataset title | Dataset URL | Database and Identifier |
|---|---|---|---|---|
| Cheng Y, Tian F, Geschwind D, He Z | 2022 | Characterization of chromatin accessibility changes in retinal ganglion cells (RGCs) following optic nerve crush | https://www.ncbi.nlm.nih.gov/geo/query/acc.cgi?acc=GSE184547 | NCBI Gene Expression Omnibus, GSE184547 |
| Cheng Y, Tian F, Geschwind D, He Z | 2022 | Core Transcription Programs Controlling Injury-Induced Neurodegeneration of Retinal Ganglion Cells | https://www.ncbi.nlm.nih.gov/geo/query/acc.cgi?acc=GSE190667 | NCBI Gene Expression Omnibus, GSE190667 |
| Syc-Mazurek SB, Yang HS, Marola OJ, Howell GR, Libby RT | 2022 | Transcriptional control of retinal ganglion cell death after axonal injury | https://www.ncbi.nlm.nih.gov/geo/query/acc.cgi?acc=GSE168789 | NCBI Gene Expression Omnibus, GSE168789 |
| Tran NM, Shekhar K, Whitney IE, Jacobi A, Benhar I, Hong G, Yan W, Adiconis X, Arnold ME, Lee JM, Levin JZ, Lin D, Wang C, Lieber CM, Regev A, He Z, Sanes JR | 2019 | Single-cell profiles of retinal neurons differing in resilience to injury reveal neuroprotective genes | https://www.ncbi.nlm.nih.gov/geo/query/acc.cgi?acc=GSE137400 | NCBI Gene Expression Omnibus, GSE137400 |

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
