## [Editor Report · eLife Assessment]

This study presents a **valuable** finding about the role of Perk (Protein kinase RNA-like endoplasmic reticulum kinase) and Atf4 (Activating Transcription Factor-4) in the integrated neurodegenerative and regenerative responses following the optic nerve injury. The authors present **solid** evidence, combining newly generated transcriptomic data with publicly available datasets to strengthen their findings. Despite some limitations in data quality and interpretation, the study is likely to be of interest to researchers studying optic neuropathies and axonal regeneration.

---

## [Referee Report · Reviewer #1 (Public review)]

Somasundaram and colleagues explore the role of transcription factors in retinal ganglion cell (RGC) death and axonal regeneration after a disease relevant insult (mechanical axonal injury). The work significantly extends our knowledge of the role of MAPK and integrated stress response (ISR) in controlling RGC fate after injury. Specifically, the manuscript shows that after axonal injury PERK-activated ISR acts through Atf4 to drive a prodeath transcriptional response in RGCs, in part by crosstalk with the prodeath JUN transcriptional program. Also, and perhaps most interesting, the work shows that PERK-ATF4 pathway activation is pro-regenerative for RGC axons. A major plus of the manuscript is that many new RNA-seq datasets are generated that describe the major prodegenerative and proregenerative gene networks altered after axonal injury. A limitation of the study is that it does not directly compare the effect of inhibiting the PERK-ATF4 pathway with inhibiting JUN and/or JUN-CHOP double deficient animals. It would also be useful, for the cell survival experiments shown in Figure 1, to examine a longer time point than 14 days to understand the long-term consequence of manipulating the PERK-ATF4 pathway.

---

## [Referee Report · Reviewer #2 (Public review)]

This manuscript investigates the role of Perk (Protein kinase RNA-like endoplasmic reticulum kinase) and Atf4 (Activating Transcription Factor-4) in neurodegenerative and regenerative responses following optic nerve injury. The authors employed conditional knockout mice to examine the impact of the Perk/Atf4 pathway on transcriptional responses, with a particular focus on canonical Atf4 target genes and the involvement of C/ebp homologous protein (Chop).

The study demonstrates that Perk primarily operates through Atf4 to stimulate both pro-apoptotic and pro-regenerative responses after optic nerve injury. This Perk/Atf4-dependent response encompasses canonical Atf4 target genes and limited contributions from Chop, exhibiting overlap with c-Jun-dependent transcription. Consequently, the Perk/Atf4 pathway appears crucial for coordinating neurodegenerative and regenerative responses to central nervous system (CNS) axon injury. Additionally, the authors observed that neuronal knockout of Atf4 mimics the neuroprotection resulting from Perk deficiency. Moreover, Perk or Atf4 knockout hinders optic axon regeneration facilitated by the deletion of the tumor suppressor Pten.

These findings contrast with the transcriptional and functional outcomes reported for CRISPR targeting of Atf4 or Chop, revealing a vital role for the Perk/Atf4 pathway in orchestrating neurodegenerative and regenerative responses to CNS axon injury.

However, the main concern is the overall data quality, which appears to be suboptimal. The transfection efficiency of AAV2-hSyn1-mTagBFP2-ires-Cre used in this study does not seem highly effective, as evidenced by the data presented in Supplementary Figure 1. The manuscript also contains several inconsistencies and a mix of methods in data collection, analysis, and interpretation, such as the labeling and quantification of RGCs and the combination of bulk and single-cell sequencing results.

Despite these limitations, the study offers valuable insights into the role of the Perk/Atf4 pathway in determining neuronal fate after axon injury, emphasizing the significance of understanding the molecular mechanisms that govern neuronal survival and regeneration. This knowledge could potentially inform the development of targeted therapies to promote neuroprotection and CNS repair following injury.

---

## [Author Response]

The following is the authors’ response to the original reviews.

**Reviewer #1 (Public Review)**:A limitation of the study is that it does not directly compare the e4ect of inhibiting the PERKATF4 pathway with inhibiting JUN and/or JUN-CHOP double deficient animals. It would also be useful, for the cell survival experiments shown in Figure 1, to examine a longer time point than 14 days to understand the long-term consequence of manipulating the PERK-ATF4 pathway.

We appreciate that both suggestions are fantastic ideas for future studies but consider them to be beyond the scope of this investigation.

**Reviewer #2 (Public Review):**
However, the main concern is the overall data quality, which appears to be suboptimal. The transfection e4iciency of AAV2-hSyn1-mTagBFP2-ires-Cre used in this study does not seem highly e4ective, as evidenced by the data presented in Supplementary Figure 1.

We appreciate the importance of the transfection efficiency of AAV2-hSyn1-mTagBFP2-ires-Cre to the interpretations of our results and acknowledge that the imaging and color schemes used required improvement. We have now validated widespread knockout in RGCs using AAV2-*hSyn1*-mTagBFP2-ires-Cre, improving the staining and imaging of LSL.tdTomato Cre reporter mice (Figure 1—figure supplement 1 and Figure 1—figure supplement 2) and using RNAScope to validate the disruption of ATF4 and CHOP, respectively, in the RGCs of ATF4 cKO and CHOP cKO mice (Figure 1—figure supplement 3 and Figure 1—figure supplement 4). Additional validation of functional knockout of these transcription factors is provided by reduction of RGC-autonomous expression of transcripts that we identified in this study to be injury-regulated in an ATF4-dependent (*Chac1*, *Atf3*, Figure 4C-E) or ATF4- and CHOP-dependent manner (*Ecel1*, *Avil*, Figure 4 and Figure 4—figure supplement 2).

The manuscript also contains several inconsistencies and a mix of methods in data collection, analysis, and interpretation, such as the labeling and quantification of RGCs and the combination of bulk and single-cell sequencing results.

Regarding the use and comparison of bulk-seq and scRNA-seq data, it is our sense that these innovative approaches will be among the impactful aspects of this study. Numerous transcriptomic studies of the optic nerve crush model exist, though it has been unclear whether major and minor technical differences would preclude deriving insights across studies without the expense and time of exact reproduction. One goal of this study was to evaluate the hypothesis that, despite the obvious limitation that RGCs represent fewer than 1% of cells in whole retina bulk transcriptomics approach, the signals amongst top differentially expressed genes (DEGs) would be dominated by injury-induced changes within RGCs and that the most robust of these changes would be readily detected across techniques and labs, serving as a cornerstone for interpreting similarities and differences in findings. We believe that the results validate this approach. Important insights gained in this study from these cross-study and cross-platform analyses include:

(1) Genes that we identify in this study as neuronal ATF4-dependent by whole retina transcriptomics include many of the most robust genes expression changes observed across multiple studies that enrich for RGCs and those that only report RGC-autonomous expression changes by scRNA-seq. This observation predicts that many of the ATF4-dependent expression changes that we report are RGC autonomous, which we further validate in this revision by RNAScope.

(2) Similarly designed whole transcriptomics studies across labs can be remarkably robust for top DEGs, showing striking similarity that allows for meaningful insights and testable hypotheses across di;erent knockout and conditional knockout mice.

(3) scRNA-seq of RGCs and bulk sequencing of FACS-enriched RGCs, unsurprisingly results in higher sensitivity for injury-induced expression changes, but the high degree of similarity that we demonstrate between the top DEGs from those studies and whole retina transcriptomics studies allows for confident inferences regarding the expected cell autonomy of reported expression changes in this model, using available resources such as the Single Cell Portal, without the expense and technical optimization required for extensive spatial transcriptomics across numerous mouse models.

Other revisions

In addition to these updates to address the public reviews, we are grateful for the reviewers’ additional recommendations and provide these further revisions:

(1) We appreciate the request to clarify with a schematic the differences between our study and a previous report (Tian *et al*., 2022). A second Correction to that study was published in July 2024, resulting in changes to the logFC values used in our original cross-study comparison and adjustments to multiple figures and tables related to the proposed transcriptional programs of ATF4, CHOP, and the other purported core transcription factors. We have therefore updated our Figure 4—figure supplement 4 and Figure 4—figure supplement 6 in accordance with that Correction to better reflect the underlying data of that study. These changes do not alter our original conclusions that: (a) both the whole retina transcriptomics approach of our study and the FACS-enriched RGC approach of that study readily detect the strong upregulation of many known ATF4 target genes after optic nerve crush (Figure 4—figure supplement 4); and (b) there are striking differences in the ATF4- and CHOP-dependent transcripts suggested by our cKO data and those suggested by the reported gRNA data. Though we had hoped that the Correction would allow us in this revision to diagram those findings and model for comparison to these cKO findings, documenting those changes and their impacts on the proposed model is beyond the scope of this study.

(2) We agree that the discordance between the gene and protein names for *Ddit3*/CHOP and *Eif2ak3*/PERK represents a challenge for clarity, even when gene names are carefully selected when referring to genes or transcripts and protein names when referring to proteins. We have therefore attempted to streamline the naming throughout, using where possible both names.